# Nitrogen deposition and climate drive plant nitrogen uptake while soil factors drive nitrogen use efficiency in terrestrial ecosystems

Helena Vallicrosa[1,2,3], Katrin Fleischer[4,5], Manuel Delgado-Baquerizo[6], Marcos Fernández-Martínez[7], Jakub Černý[8], Di Tian[9], Angeliki Kourmouli[10,11], Carolina Mayoral[11,12], Diego Grados[13], Mingzhen Lu[14,15], César Terrer[1]

**1** Department of Civil and Environmental Engineering, Massachusetts Institute of Technology, Cambridge, Massachusetts, USA.

**2** Community Ecology Unit, Swiss Federal Institute for Forest, Snow and Landscape Research WSL, CH-8903 Birmensdorf, Switzerland

**3** Plant Ecology Research Laboratory PERL, School of Architecture, Civil and Environmental Engineering ENAC, EPFL, CH-1015 Lausanne, Switzerland

**4** Department of Biogeochemical Signals, Max-Planck-Institute for Biogeochemistry, Jena, Germany.

**5** Section Systems Ecology, Amsterdam Institute for Life and Environment, Vrije Universiteit Amsterdam, The Netherlands

**6** Laboratorio de Biodiversidad y Funcionamiento Ecosistémico, Instituto de Recursos Naturales y Agrobiología de Sevilla (IRNAS), CSIC, Sevilla, Spain

**7** CREAF, E08193 Bellaterra (Cerdanyola del Vallès), Catalonia, Spain.

**8** Forestry and Game Management Research Institute, Strnady 136, Jíloviště 252 02, Czech Republic.

**9** State Key Laboratory of Efficient Production of Forest Resources, Beijing Forestry University, Beijing 100083, China

**10** Lancaster Environment Centre, Lancaster University, Lancaster, UK

**11** Birmingham Institute of Forest Research, University of Birmingham, United Kingdom

**12** School of Biosciences, Edgbaston Campus, University of Birmingham, United Kingdom

**13** Department of Agroecology, Climate and Water, Aarhus University, 8830 Tjele, Denmark

**14** Department of Environmental Studies, New York University, New York, NY 10012, USA.

**15** Santa Fe Institute, New Mexico, NM 87501, USA

Correspondence to: Helena Vallicrosa (helena.vallicrosa@gmail.com)

ABSTRACT

The role of plants in sequestering carbon is a critical component in mitigating climate change. A key aspect of this role involves plant nitrogen (N) uptake (Nup) and N use efficiency (NUE), as these factors directly influence the capacity of plants to store carbon.

However, the additive contribution of N deposition, soil factors (biotic and abiotic) and
climate to the plant N cycle remains inadequately understood, introducing significant
uncertainties into climate change projections. Here, we used ground-based observations
across 159 field experiments (including above and belowground information) to calculate
Nup and NUE and identify their main drivers in natural ecosystems. We found that global
plant Nup is primarily driven by N deposition, mean temperature and precipitation, with
Nup increasing in warmer and wetter areas. In contrast, NUE is driven by soil biotic and
abiotic factors. Specifically, NUE decreased with the intensity of colonization by arbuscular
mycorrhizal fungi and increased with soil pH and soil microbial stocks. Nup and NUE
presented opposite latitudinal distributions, with Nup higher on tropical latitudes and NUE
higher towards the poles. Total soil N stocks were not found to be a driver of Nup or NUE.
We also compared our results with TRENDY models and found that models may
overestimate Nup by ~ 100 Tg N yr$^{-1}$ in the tropics and triple the standard deviation at boreal
latitudes. Our findings emphasize the effect of N deposition and soil microbes that, in
addition to climate and soil pH, are crucial for accurately predicting ecosystems' capacity to
sequester carbon and mitigate climate change at a global scale.
Plain language summary
We used field empirical data worldwide to calculate plant nitrogen uptake (Nup) and
nitrogen use efficiency (NUE) in woodlands and grasslands and determine their drivers,
which can be used as empirical validation for models. Even though some regions of the
world have decreased their N deposition, N deposition is still the most important driver
explaining plant nitrogen uptake, aside from climatic variables.

# 1.Introduction

Climate and nutrient availability play significant roles in the capacity of plants to sequester carbon (C). Nitrogen uptake (Nup) and nitrogen use efficiency (NUE) are fundamental processes in plant-soil N cycling, which in turn impact biodiversity, ecosystem productivity, C sequestration, food security, and human health (Peñuelas et al., 2020). Hence, realistic quantifications of Nup and NUE and the understanding of their drivers are crucial to predicting the fate of terrestrial ecosystems under a changing environment. Climate, biomass production, and Nup are strongly intertwined, where hotter and wetter ecosystems have the capacity to grow more, increase their N demand, and therefore absorb more N if available (Berntson et al., 1998; Wu et al., 2011). Nonetheless, several factors can affect N availability. Traditionally, total soil N stocks were used to proxy N availability or plant Nup. Although this correlation is weak, it is still used from a modeling perspective (Stevens et al., 2015; Vicca et al., 2018) assuming that total soil N positively correlates with N availability.

The soil community (i.e. microbes and fungi) plays a crucial role in global biogeochemical cycles governing processes such as N fixation, nitrification, denitrification, and general organic matter and nutrient turnover (Aber et al., 2001; Sinsabaugh et al., 2002; Sinsabaugh et al., 2008; Crowther et al., 2019; Delgado-Baquerizo et al., 2020). In turn, the soil community can also act as a buffer in case of nutrient excess (Wall et al., 2015) or contribute to nutrient foraging in case of nutrient deficiencies (Chen et al., 2018), shaping ecosystem functioning (Bardgett and van der Putten, 2014). Therefore, the soil community is expected to interfere substantially with plant Nup and NUE. N deposition is another agent relevant for global N cycles, which has increased from ~30 to ~80 Tg N/year worldwide since 1850 (Kanakidou et al., 2016) with an associated increase in N availability (Elser et al., 2010; Battye et al., 2017; Peñuelas et al., 2020). Consequently, reliable quantifications of plant Nup and NUE need to include climatic factors as well as soil biotic factors and N deposition.

N regulates the capacity of ecosystems to store C (Hungate et al., 2003; Fernández-Martínez et al., 2014; 2019; Wang et al., 2017) and respond to climate change drivers (Fleischer et al., 2019; Terrer et al., 2019; Walker et al., 2021; Zhou et al., 2022) being the C-N assembly relevant for land surface models (LSM). Eight of the LSM of the TRENDY ensemble v8 (Sitch et al., 2015), a model ensemble designed to disentangle the effects of climate, CO2, land use, and land cover change, include representations of the N cycle and plant Nup. Nonetheless, their parameterization of N cycling is poorly constrained by observations (Zaehle et al., 2014; Fowler et al., 2015; Braghiere et al., 2022). Consequently, when models are assembled, the result leads to accumulated uncertainty (Prentice et al., 2015; Franklin et al., 2020) and therefore divergent predictions of the land sink (Zaehle et al., 2014; Stocker et al., 2016; Arora et al., 2020). Furthermore, when accounting for N interactions, LSMs do not generally consider the direct effects of microorganisms, missing out on the role of soil bacteria or mycorrhizae on plant nutrient uptake. Including global calculations of plant Nup and NUE based on empirical data, as well as accounting for climate, N deposition, and soil biomass interactions, would potentially refine the N accountability in LSM.


Here, we gathered information from 159 plots worldwide that describe woodlands and
grasslands across different biomes to calculate plot-based plant Nup and plant NUE using
exclusively empirical field data. Our analyses combine N concentration and net primary
productivity (NPP) data in different aboveground and belowground plant tissues (i.e.,
leaves, roots and stem). We used linear models to identify the drivers of Nup and NUE,
including N deposition, soil microbes, woodiness, and climatic factors. We then upscaled
those results using machine-learning models to quantify yearly plant Nup and plant NUE at
a global scale in natural terrestrial ecosystems (woodlands and grasslands) and compared
these results with simulations from LSM. We hypothesize that factors such as N deposition
and soil microorganisms have significant impacts on Nup and NUE respectively, playing a
role as important as climatic drivers. We expect the ground-based data, and incorporation
of these N-relevant drivers to increase the accuracy of global Nup quantifications. Thus, a
mismatch between our estimation and current TRENDY simulation outputs is expected.

**2. Results and discussion**
**2.1 Nitrogen uptake and nitrogen use efficiency**
Our findings indicate that N deposition and climate are fundamental factors explaining plant
Nup on a global scale (Fig. 1). We found a positive relationship between Nup and
accumulated N deposition, mean annual temperature (MAT), and mean annual
precipitation (MAP). Thus, regions that are warm, wet, and with higher levels of N
deposition exhibit the highest rates of Nup. Our results show that N deposition is strongly
contributing to fulfilling the N demand in productive environments, alleviating potential N
limitations, and allowing high plant Nup levels. On the other hand, our empirical results did
not show important relationships between plant Nup and soil microbial interactions nor soil
physico-chemical variables (Fig. 1a). Those results include no significant relationship
between soil N stocks and Nup at a global scale (Fig. S1a), which has been further tested
following a univariate approach. This result discourages the use of soil N to infer N
availability.

**Plant nitrogen uptake**

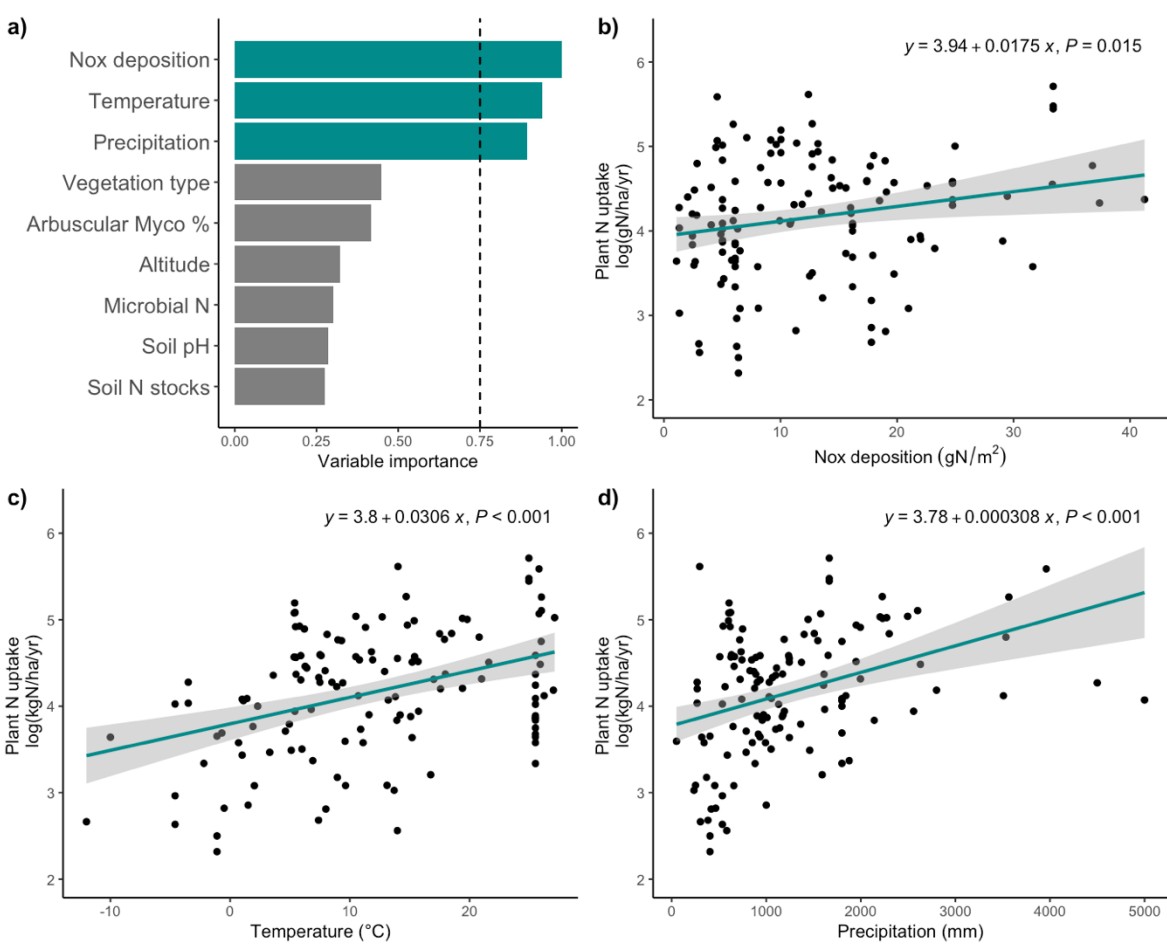

*Figure 1. a) Variable importance plot for the general linear model (GLM) describing plant nitrogen uptake (Nup). The*
*dashed line is set at 0.75, separating the threshold for important variables. The GLM model pseudoR$^2$ was 0.349. Linear*
*regressions were displayed describing plants' nitrogen and important variables b) accumulated Nox deposition from 1901*
*to 2021, c) mean annual temperature, and d) mean annual precipitation. Equation and p-value per regression are*
*displayed. Acronyms: Nox: oxidized nitrogen, N: nitrogen, Myco %: Mycorrhizal percentage.*
In contrast, our model selection analysis identified soil biotic and abiotic factors as the main
NUE drivers (Fig. 2). We found that NUE decreased with AM % and increased with soil pH
and soil microbial N stocks. Thus, plant species prone to be colonized by arbuscular
mycorrhizae, are less efficient in N use to build biomass. In contrast, basic pH and abundant
soil microbial stocks facilitate higher NUE rates. Even though soil variables appear to be
important for NUE, soil N stocks remain unrelated to NUE in the model and when tested
individually (Fig. S1b). Despite climatic variables such as MAT and MAP not appearing as
important variables explaining NUE, they are to some extent represented in the soil
variables. As shown in Table S2, soil variables are not independent of climatic variables,
since they have some degree of correlation.

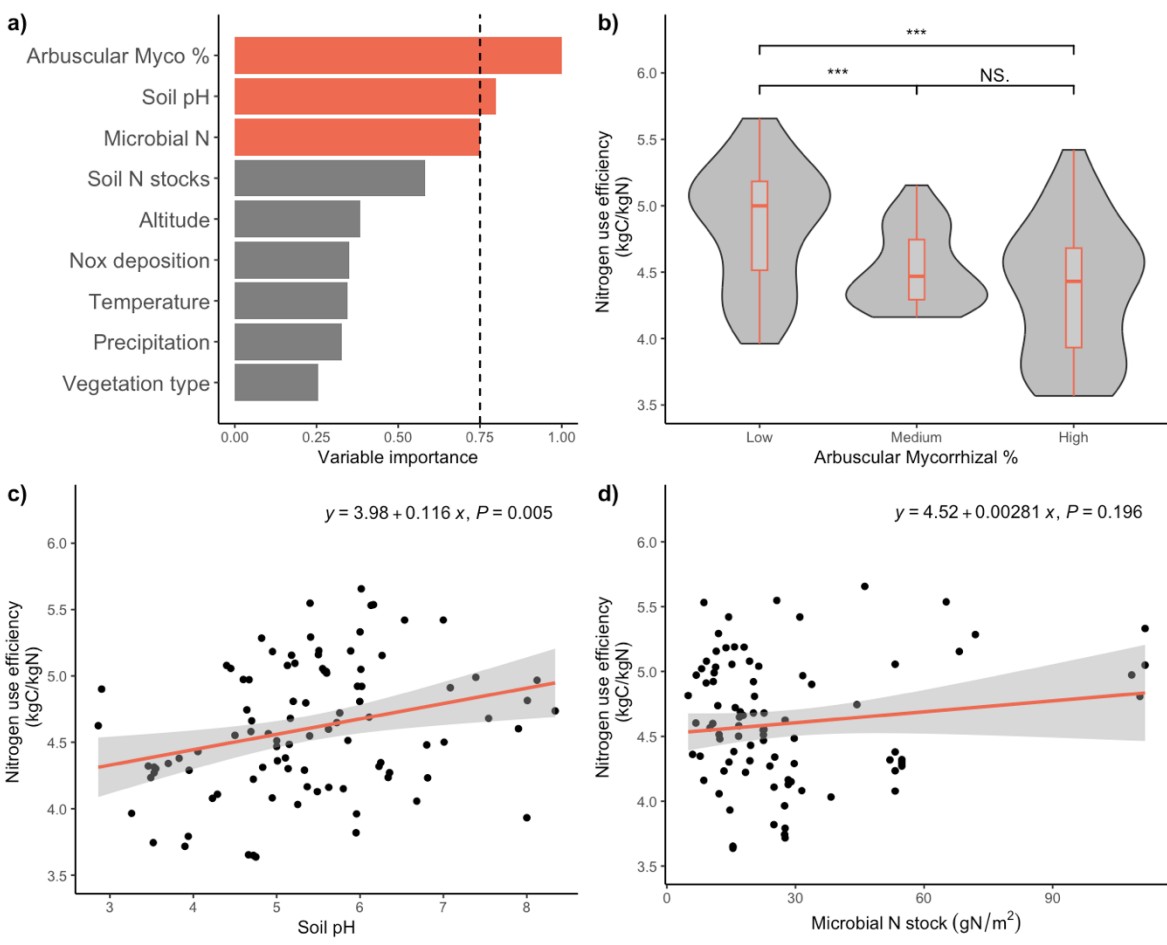

Figure 2. a) Variable importance plot for the generalized linear model describing nitrogen use efficiency (NUE). The model preudoR2 was 0.355. The dashed line is set at 0.75, separating the threshold for important variables. In b) arbuscular mycorrhizae percentage is divided into low, medium, and high, and NUE is displayed. * = P-value < 0.05, ** = P-value < 0.01, *** = P-value < 0.001. Linear regressions were displayed describing plants' nitrogen use efficiency c) soil pH and d) microbial N stocks. Equation and p-value per regression are displayed.

## 2.2 Global maps of Nup and NUE

Next, we used a machine-learning model, XGBoost, to estimate the global magnitude and distribution of Nup and NUE by extrapolating the site-level relationships to the global scale. For methodological consistency, the XGBoost model was trained using the same nine variables as the linear model. The model identified temperature, precipitation, and N deposition as the most critical factors for describing Nup (Fig. S2), consistent with the linear model, though ranked in a slightly different order. Partial dependence plots further supported the relationships observed in the linear models (Fig. S3).

The upscaled Nup map showed a total yearly Nup of 842.215 ± 236.11 Tg of N, with a mean coefficient of variation of 26.77 % (Fig. S4) and an $R^2$ of 0.54 (Fig. S2). The lowest Nup values were on boreal latitudes and mountain ranges such as the Rockies (USA), Andes (South America), various European ranges, and the Himalayan plateau (Asia). In contrast, higher rates of Nup were predicted in temperate latitudes in Europe, the eastern United States,

Southeast Asia, eastern Australia, much of South America, and central Africa. The highest values were found in the Congo region, where high N deposition, temperature and precipitation converge (Fig. 3a). The Nup map shows a strong NPP influence, driven by temperature and precipitation, but including N deposition.

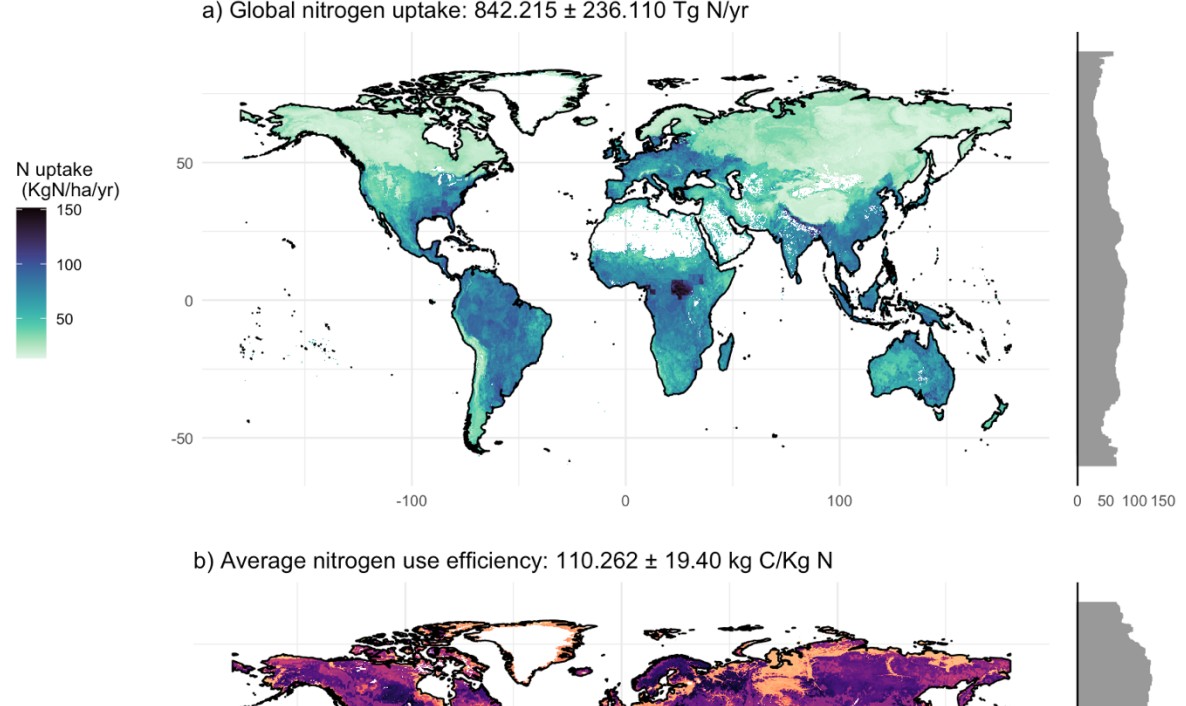

a) Global nitrogen uptake: 842.215 ± 236.110 Tg N/yr

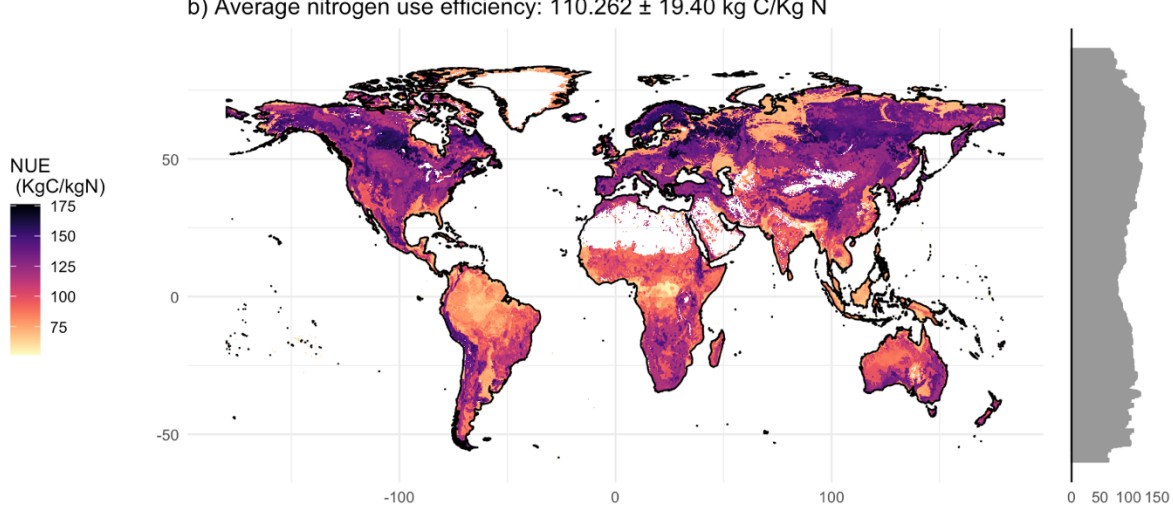

b) Average nitrogen use efficiency: 110.262 ± 19.40 kg C/Kg N

*Figure 3. Upscaled global maps describing a) Plant nitrogen uptake and b) Nitrogen use efficiency. The total amount of nitrogen uptake calculated per year is 842.215 Tg of N with a standard deviation of ± 236.11. The mean value of global nitrogen use efficiency is 110.26 kg of C per kg of N, and its standard deviation is 19.40. White areas indicate no data due to absence of grasslands or woody vegetation.*

The machine-learning model for NUE identified microbial N stocks, altitude, precipitation, soil pH, and AM% as the most important drivers (Fig. S5). The direction of these effects matched those found in the linear model, with the addition of precipitation and altitude (Fig. S6). The global mean NUE estimate was 110.262 units of C per unit of N with a mean coefficient of variation of 17.89 % (Fig. S4) and an $R^2$ of 0.44 (Fig. S5). The global map showed lower NUE around the Equator, progressively increasing towards the poles. Nonetheless,

some heterogeneous patches of high and low NUE appeared between 50 and 60 degrees
latitude north (Fig. 3b).

**2.3 Global-scale Nup comparison with TRENDY models**
We compared our estimates for total yearly Nup, upscaled from field observations, with the
mean Nup across the eight N models included in TRENDY. TRENDY models simulate higher
Nup in the tropical regions, reaching differences of around 100 kg N ha$^{-1}$ yr$^{-1}$ in those areas
(Fig. 4a), representing more than 100% of the Nup estimated by field observations (Fig. 4b).
Other areas like northern and northeastern North America, Southeast Asia, and northern
Eurasia also ashow higher Nup values in TRENDY models than in field observations. In boreal
latitudes, TRENDY model deviations for Nup exceeded 300% overestimation. On the other
hand, areas where the machine-learning models predict higher values than TRENDY models
include the southern latitudes. Middle Eastern regions, the Somali peninsula, and the Rocky
Mountains (Fig. 4). Overall, TRENDY models estimate Nup values that are higher by 16.61
kg N ha$^{-1}$ yr$^{-1}$ on average, accounting for 48.54 % of the variability. When aggregating total
yearly Nup, LPX-Bern and CLM5.0 were the models that predicted overall values exceeding
our confidence range, suggesting significantly higher Nup (Fig. S7).

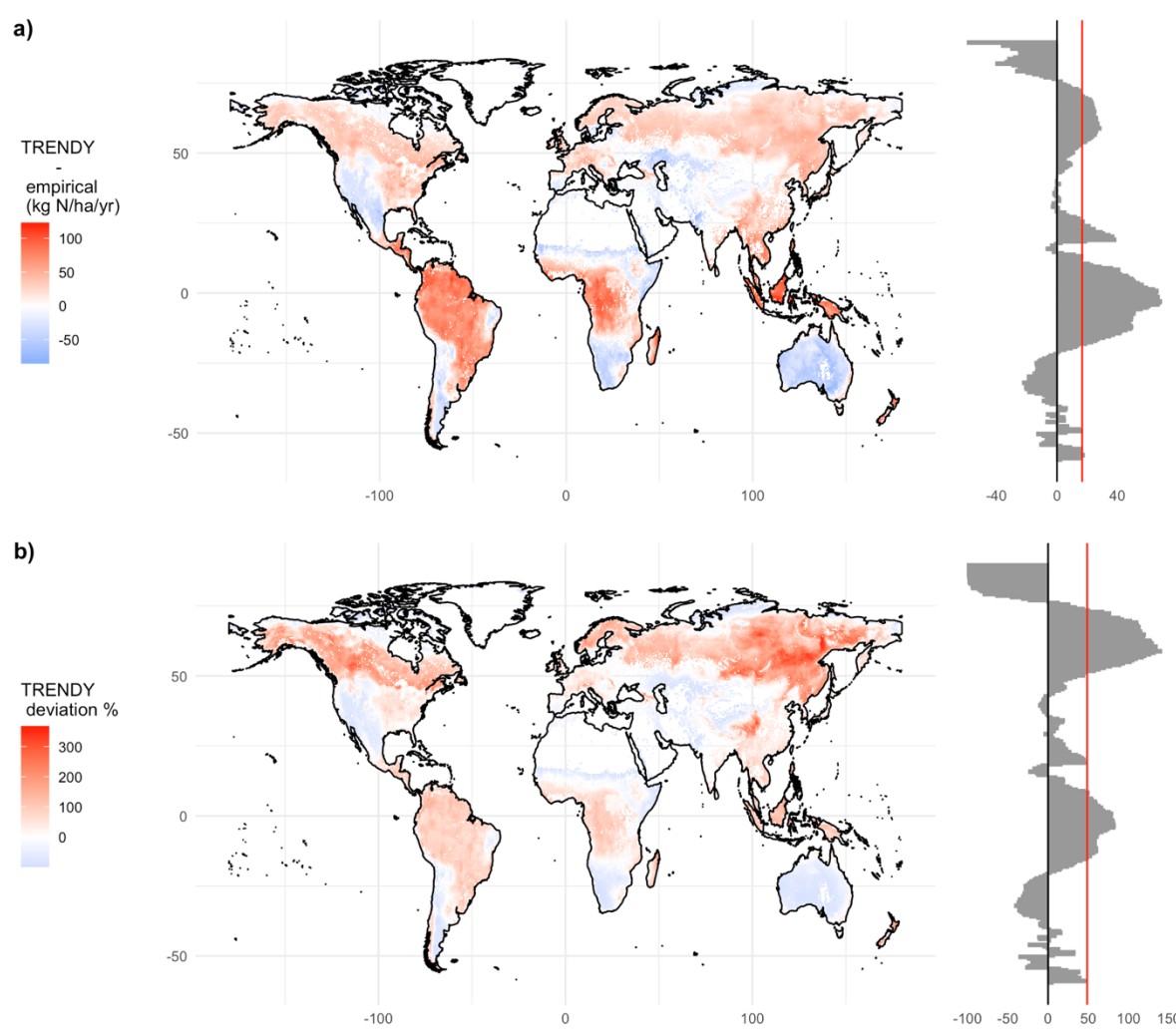

*Figure 4. Comparison between the mean of the nitrogen uptake provided by TRENDY v8 models minus the upscaled*
*nitrogen uptake. The red color stands for higher values on the TRENDY model, and the blue color stands for higher nitrogen*
*uptake values on the upscaled approach. In a) units in kg N ha$^{-1}$ yr$^{-1}$ and in b) units in percentage of deviation from field*
*upscaling. Latitudinal aggregation on the right, with a red vertical line showing a ) the mean of the total comparison at*
*16.61 kg N ha$^{-1}$ yr-1 and b) the mean percentage of deviation at 48.54%.*

## 2.4 Nup global drivers and implications

Our models estimated annual global plant Nup at 842 ± 236 Tg of N. This figure is consistent
with the findings of Peng et al., 2023, which estimated 950 ± 260 Tg of N, and Braghiere et
al., 2022, with an estimated uptake of 841.8 Tg N. The slight variations can be attributed to
differences in methodologies and data sources (simultaneous plot-averaged records vs
individual-level records) used in these studies. In our study, linear models and machine
learning models consistently identify N deposition, temperature, and precipitation as global
drivers of Nup. Hotter and wetter environments increase biological activity, leading to more
biomass production and therefore more N demand. An increase in N demand with enough
N availability is associated with an increase in Nup. The accumulation of N deposition
throughout time originating from anthropogenic sources has been increasing the N
availability in some areas, generally close to industrial or agroforestry pools. Hence, in a
global change context where $CO_2$ fertilization and temperature increase have generated a

greening effect (Ruehr et al., 2023), areas with higher N deposition were able to better supply the increasing N demand. Thus, according to our results, anthropogenic N supply may now be as important a driver of Nup as climate.

These results are concerning since our data emphasize the far-reaching influence of human-induced N deposition in shaping global Nup patterns. Some regions such as Europe, the eastern USA, and the tropics have decreased their N deposition levels during the last four decades (Ackerman et al., 2019). Nonetheless, these reductions have not yet translated into measurable changes in how N deposition affects natural woodlands and grasslands, which still appear to be primarily driven by N deposition. This sustained input of anthropogenic N has been associated with a fertilization effect, enhancing the land C sink by 0.72 Pg C yr$^{-1}$ during the 2010s (Gurmesa et al., 2022). However, this N fertilization effect has showed signs of saturation in forests and grasslands (Tian et al., 2016; Peng et al., 2020), where increases in biomass production (and therefore C sink strength) have slowed. Consequently, this excess N input from N deposition may no longer be captured by biomass and instead contributes to N leaching, eutrophication, acidification, loss of biodiversity, and $N_2O$ emissions (Aber et al., 1989; Gundersen et al., 1998; Bobbink et al., 2010), exacerbating environmental problems.

**2.5 NUE global drivers and implications**

Our results predict a mean NUE of 110 ± 19 kg C per kg N, driven by soil biotic and abiotic factors. The main divergence between linear models and machine learning models is the importance of altitude and precipitation, which showed explicit relevance only in the machine learning model. We attribute these differences to the nature of the models, where machine-learning models accommodate correlations without modifying their variable importance. Thus, the important variables in the linear model could also have embedded important latitudinal gradients and therefore altitudinal or precipitation gradients. Our NUE predictions differ from Peng et al., 2023, who reported a mean NUE of 76 ± 26 kg C per kg of N. The main distinction is that our approach included biotic factors, such as mycorrhizal associations and microbial interactions, which explained NUE better than abiotic factors. In contrast, Peng et al., 2023 based their estimates solely on abiotic factors. We do not consider environmental variables such as precipitation to be totally independent from NUE relations, as they influence important biotic variables such as AM %, and microbial N stocks. Nonetheless, the results showed that including biotic variables may result in more efficient use of N by plants at a global scale.

The response of NUE has been postulated as a method to assess N saturation in plant communities (Shcherbak et al., 2014). A negative relationship between N addition and NUE, along with lower NUE levels, can indicate N saturation (Iversen et al., 2010). In our study, tropical areas exhibit the lowest NUE, suggesting lower N limitation, consistent with previous global upscaling studies using different methods (Du et al., 2020; Vallicrosa et al., 2022). According to the soil age hypothesis (Walker and Syers, 1976), N accumulates in ecosystems over time through biological processes. Thus, younger ecosystems, such as those at high elevations or with lower pH, show higher values of NUE and are likely to be

more N-limited. Our results show only a modest effect of N saturation due to N deposition, so further studies are needed to better assess where and under what circumstances areas are N saturated due to N deposition at a global scale.

Biological activity, particularly the type of mycorrhizal associations and soil microbial N stocks, had a strong impact on N dynamics. Arbuscular mycorrhizal associations, which dominate tropical ecosystems (Soudzilovskaia et al., 2019), are hypothesized to be more efficient in nutrient acquisition and more abundant in areas with fast N cycling (Averill et al., 2019). Our models show that AM associations are linked to lower NUE, possibly due to abundant N and the high efficiency of AM fungi in acquiring it. Conversely, N acquisition appears more efficient in regions with higher microbial N stocks. As described by Kuzyakov and Xu (2013), we hypothesize a potential competition effect between soil microbes and plants for N, but further studies are needed to corroborate this mechanism. Given the central role of biological processes in regulating N transformation and uptake, it is reasonable to conclude that total soil N stocks (which include all forms and pools of N) are not reliable indicators of N availability or plant Nup.

## 2.6 Latitudinal discrepancies between Nup map and TRENDY

LPX-Bern and CLM5.0 models projected Nup values significantly above our estimates, reporting 1471 and 1454 Tg N/yr, respectively (Fig. S7). Although the average of all TRENDY models falls within our Nup confidence range, spatial discrepancies are large. In tropical and northern latitudes, TRENDY models predicted higher Nup than our estimates. In contrast, TRENDY models predicted lower Nup values in southern latitudes, western Asia, and the Rocky Mountains. This mismatch could result in an overestimation of the terrestrial C sink and a misinterpretation of the role of vegetation in N cycling. One possible explanation is that LSMs overestimate biomass production by failing to account for growth-limiting factors such as phosphorus limitation, drought, or biotic competition. Alternatively, overestimation of tissue N concentration could also lead to inflated Nup values, which would inherently result in lower modeled NUE. In our calculations, we explicitly accounted for the variability if N concentration and net primary productivity among tissues, including leaf resorption, to generate more accurate Nup and NUE estimates.

## 2.7 Representativity and future research

An inherent challenge in ecological studies of this scale is to ensure the global representativeness of the dataset since systematic geographical sampling biases underrepresent the global South (Auge et al., 2024). In this study, 28 % of the data comes from areas below 15º latitude, outside the US, Europe, or China (Fig. S8). In terms of ecosystem representation, the Whittaker diagram shows coverage across all biomes (Fig. S9), with subtropical deserts, tundra, and temperate rainforest being the least represented. Nonetheless, we acknowledge that calculations based on empirical data, especially when a portion of the data has undergone a gap-filling process, may still carry biases related to sampling and upscaling. These biases are primarily shaped by the biomes with higher observational density. Still, we argue that calibrating and validating models built on mathematical assumptions using field measurements is essential to better anchor model

outputs to reality. Global scale approaches like this one aim to provide broad estimates of
planetary processes, accepting a scale-precision tradeoff. As such, we do not recommend
applying our approach to fine-scale predictions, since local heterogeneity, including forest
type, species composition, and land management practices may not be captured accurately.
This study is focused on a quantitative approach at a global scale, attempting to target
variables' relative importance on Nup and NUE along with its correlations to environmental
and biotic variables. In future research, specific data detailing the different N fractions
obtained at a global scale (e.g. organic-inorganic, ammonium-nitrate) and a more
mechanistic framework are strongly encouraged. We also encourage empirical studies
targeting underrepresented biomes, especially from the global South. Approaches such as
those in Niu et al., 2016 quantifying the fraction of Nup taken by plants, leached and
retained in the soil at a global scale are crucial to enhancing our understanding of the N
cycle and its interactions with ecosystems.
**3. Conclusion**
We found that N deposition and climatic variables are the main global-scale factors
explaining Nup. Regions that are warm and wet, and subject to higher levels of N deposition,
exhibit the highest rates of Nup. This result highlights the far-reaching influence of
anthropogenic N deposition in shaping global Nup patterns. Interestingly, NUE was shown
to be driven by soil biotic and abiotic factors, emphasizing the importance of soil
microorganisms and pH as regulators of the N cycle. We further demonstrated that total
soil N stocks are not significant drivers of either Nup or NUE. Our upscaling results showed
large spatially explicit discrepancies compared to TRENDY Nup values, with TRENDY
simulating higher absolute values in tropical regions and larger deviations in boreal
latitudes. This spatial mismatch between empirical data and land system models could
substantially affect model accuracy and future projections of the C sink, particularly if
tropical C storage capacity has been overestimated. Our findings offer key insights for
improving understanding of $C - N$ interactions, N cycling, and N uptake in terrestrial
ecosystems, and they underscore that N deposition remains a dominant global driver of
plant Nup.
**4. Methods**
**4.1 Data extraction**
We gathered 159 field plot data representing 129 different sites in natural conditions
coming from published data or repositories (Table S1). The data included information on
the dominant species and vegetation type (grassland, coniferous, or broadleaved), foliar
and root N concentration, foliar and root biomass production, and stem biomass production
in the case of woody plants at the same location and time. In situ measurements for foliage
and fine roots are the most relevant for Nup calculation (Dybzinski et al., 2024), so all our
data points include biomass production (NPP) and N content (N%) of leaves and roots. We
gathered 45 data points, representing 28% of the data, coming from latitudes under the 15º
latitude, despite the systematic lack of field sampling on some regions of the earth such as
the global south. We also complemented the dataset with field values of litter biomass

production, litter N concentration, stem N concentration, soil pH, soil C %, soil N %, soil texture, soil moisture, mean annual precipitation, mean annual air temperature, and altitude. We included woody and grassland natural environments (Fig. S8), including representation from most biomes according to Whitakker's diagram (Fig. S9). Each data point covered by the analysis has been collected from 1984 to 2022. If stem N was missing, happening in 25% of the data entries, we gap-filled it with the mean value of its vegetation type (coniferous=0.33 or broadleaved=0.52%). With leaves, stem, and roots we calculated the gross Nup (see in the next section). By subtracting the amount of N recovered during leaf senescence we obtain the net Nup. If litter biomass was missing, 52% of the time, we assumed it to be the same amount of green leaf biomass production. If litter N concentration was missing, we calculated the net Nup using the predicted value from a linear model created with net Nup in the base of gross Nup, in 33% of the entries. This model had an r2 of 0.88, a p-value < 2.2e-16, and a correlation of 0.72 between gross and net Nup.

We extracted mean annual precipitation from WorldClim2 (Fick and Hijmans, 2017), as well as soil pH, soil C, and soil N, soil moisture, soil bulk density, and soil texture from soilGrids (Poggio et al., 2021). All soil data for the topsoil layer (0-15 cm). We also identified the potential mycorrhizal association from the dominant species (% of colonization) based on Soudzilovskaia et al. 2020, and categorized it into 0, 50, or 100 arbuscular mycorrhizal (AM) percentages, since AM is the most abundant and common throughout the globe. When dominant species were not provided, we extracted the AM% of colonization based on the AM map of Soudzilovskaia et al. 2019 and the coordinates of our samples. Moreover, we extracted the microbial N stock from Xu et al. 2013. We calculated and obtained the accumulated oxidized N deposition from Yang and Tian, 2022 from 1901 to 2022 by georeferencing each field plot. Oxidized and reduced N deposition are correlated and are thought to have similar ecological effects (Sutton and Fowler, 1993; Yang and Tian, 2022). Oxidized forms generally come from combustion reactions while reduced forms generally come from agricultural practices. We decided to use the oxidized form because it is the most equally distributed at a global scale.

### 4.2 Nitrogen uptake and Nitrogen use efficiency calculation
We calculated the increase in annual N stock for each tissue (leaves, stem, roots, and litter) by multiplying the biomass increase by its N concentration. We obtained the gross annual Nup by aggregating tissue's Nup (roots, leaves, and stem if woody). To account for the N that has been reabsorbed before senescence, we subtracted the litter N stock from the green leaves N stock. We subtracted the reabsorbed N from the gross Nup to obtain the final net Nup value as follows:

$$Nup = (NPPleaves * Nleaves + NPPstem * Nstem + NPProots * Nroots) - (NPPleaves * Nleaves - NPPlitter * Nlitter)$$

*Nup* = Plant nitrogen uptake (kg N/ha/yr)
*NPP* = Net primary production (kg of biomass/ha/yr)
*N* = Nitrogen (% in dry weight)

We calculated the nitrogen use efficiency (NUE) by calculating the total amount of biomass
produced in leaves, stems, and root tissue divided by the amount of nitrogen in each tissue.
It will give the amount of biomass produced by a unit of nitrogen.
$$NUE = (NPP_{leaves} / Nup_{leaves}) + (NPP_{stem} / Nup_{stem}) + (NPP_{roots} / Nup_{roots})$$
*NUE* = Nitrogen use efficiency (kg C / kg N)
*NPP* = Net primary production (kg of biomass/ha/yr)
Nup = Plant nitrogen uptake (kg N/ha/yr)

**4.3 Linear statistical analysis**
Nup and NUE values correlate 34% (Fig. S10). From the available variables collected, we
selected the less correlated ones using the *cor* function in R to deal with multicollinearity.
The less correlated variables selected were mean annual air temperature, mean annual
precipitation, altitude, arbuscular mycorrhizae percentage, microbial N stock, soil N stock,
soil pH, accumulated oxidized N deposition from 1901 to 2022, and woodiness. The biggest
collinearity among variables was 0.52 between mean annual temperature and AM presence
(Table S2). With the less correlated variables, we created generalized linear models using
Nup and NUE as dependent variables. The family was set up as Gamma with an inverse link
to fulfill the residual normality requirements. We also calculated the Variance Inflation
Factor (VIF) with the *vif* function of the car R package (Fox and Weisberg, 2019) of the
aggregated model to validate VIFs lower than 4. We performed a model selection using the
*dredge* function in the MuMIn R package (Barton, 2023) and chose the best linear model
based on its lowest AIC. We calculated the variable importance using the function *sw* on the
MuMIN R package (Barton, 2023), which is a standard method based on Akaike weights
(Giam and Olden, 2016). We calculated the pseudo R square of the models using the
function pR2 from the package pscl (Jackman, 2020). Figures were created using the R
package *ggplot2 (*Wickham, 2016*)*.

**4.4 Nitrogen uptake and nitrogen use efficiency upscaling**
To upscale Nup and NUE to global grasslands and woody vegetation, we used extreme
gradient boosting (XGBoost) models splitting the database into train, test, and validation
using a standard ratio of 70:20:10, respectively (Lever et al., 2016). Extreme gradient
boosting is a machine learning algorithm that builds ensemble decision trees, applying
regularization and pruning techniques to improve performance and prevent overfitting
(Chen et al., 2016). XGBoost is a non-parametric model particularly indicated for high
performance in sample sizes above 100 data points, overcoming potential problems of
autocorrelation and optimizing predictive power. We trained an XGBoost model using the
R package *xgboost* (Chen et al., 2023), forcing an early stop based on minimum root mean
squared error to avoid overfitting and setting up the objective as a gamma regression. We
optimized the parameters based on performance (prediction $R^2$) at a maximum depth of 6,
minimum child weight of 1, and eta of 0.3, which are generally standard values. We
considered the same independent variables included in the linear model without
interactions. We repeated this process 20 times with random database separation to
stabilize the variability due to randomness in subset splitting. We extracted the variable
importance of each model using the function *xgb.plot.importance* on the *xgboost* R package
(Chen et al., 2023), calculated the mean of the values among the 20 different training sets,
and displayed it using ggplot. We calculated partial dependence plots using the function
*partial* in *purrr* R package (Wickham and Henry, 2023) to explore the non-linear relations
on the models. To calculate the model performance, we calculated the mean squared error
of the test set and the r squared of the predicted vs observed in the validation subset,
considering the validation set as completely independent.

To predict the values at a global scale, we used the spatial explicit mean annual
precipitation, mean annual temperature, and altitude variables provided by WorldClim2
(Fick and Hijmans, 2017); the microbial N stocks by Xu et al. 2013; the oxidized accumulated
N deposition from 1909 to 2022 calculated from Yang and Tian 2022 and soil N stocks and
soil pH provided by soilGrids 2.0 (Poggio et al., 2021) at 15 cm depth. We reclassified the
European Space Agency Land Cover (ESA-LC) map (Defourny, 2019) (Table S3) and we
downscaled its resolution to 2 km using the raster R package (Hijmans, 2023). We upscaled
each of the 20 Nup and NUE models using the trained XGBoost models and their prediction
per pixel at 2 km resolution and calculated the mean to obtain the final maps. We
parallelized the process using the *parallel* function and *spaDES.tools* R package (McIntire
and Chubaty 2023) to accelerate the upscaling. We masked areas not considered woodlands
or grasslands in natural conditions according to the European Space Agency cover map
(Defourny, 2019) (Table S3), and then, we obtained a map of the yearly Nup, Nup standard
deviation, and annual NUE. We obtained the final number of yearly Nup by summing all the
pixels available.

**4.5 Nitrogen uptake comparison with TRENDY models ensemble**
We obtained the available Nitrogen uptake of Vegetation (fNup) variable associated with
all the available models in TRENDY v8 S3 (Sitch et al., 2015; Le Quéré et al., 2018). The
models containing fNup are ORCHIDEE, LPX-Bern, LPJ-GUESS, JULES, JSBACH, DLEM,
CLM5.0, and Cable-POP, and the S3 experiment in the simulation considering the adaptation
of $CO_2$, land use, N deposition, and climate from 1850 representing current environmental
conditions. We calculated the yearly mean Nup from 1984 to 2022 for each model, and the
average of all of them combined, to obtain a final yearly value. Then, we calculated the
difference between each model included in the TRENDY ensemble and our Nup estimations.
After, we averaged all the fNup values in a unique spatial explicit representation and
compared it with our estimations. We also calculated the latitudinal mean of the difference
to achieve a latitudinal profile and calculated the overall mean.

**Contributions**
H.V. and C.T. conceived the project; C.T. got the funding and supervised the work; H.V.,
C.M., A.K., J.C., and D.T. collected and compiled the data; H.V. curated and analyzed the
data, created the visuals and wrote the first draft; H.V., C.T., M.D.B., M.F.M., M.L., D.G.,
contributed with substantial ideas and feedback on the manuscript; all authors revised,
edited, and agreed on the final manuscript.

**Data availability statement**

The data gathered for this study, code and produced models are available at Zenodo (Vallicrosa Pou, 2024).

**Competing interests**

The authors declare no competing interests

**Acknowledgments**

We acknowledge the members of the Terrer Lab for providing scientific consulting as well as mental and emotional support during the investigation. We acknowledge the Pioneer Center Land-CRAFT, Department of Agroecology, Aarhus University for making possible this collaboration with D.G. J.C. was supported by the National Agency of Agricultural Research of the Czech Republic (Project No. QK22020008) and the Ministry of Agriculture (CR), institutional support MZE-RO0123, and he thanks FGMRI technician staff for help with field and lab works. M.F.M was supported by the European Research Council project ERC-StG-2022-101076740 STOIKOS and a Ramón y Cajal fellowship (RYC2021-031511-I) funded by the Spanish Ministry of Science and Innovation, the NextGenerationEU program of the European Union, the Spanish plan of recovery, transformation and resilience, and the Spanish Research Agency.

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
