# Peer review of "Nitrogen deposition and climate drive plant nitrogen uptake while soil factors drive nitrogen use efficiency in terrestrial ecosystems"

_EGUsphere, 2024_

## Referee Comment (RC1)

Reviewer comments for egusphere-2024-3661

This paper sought to quantify the effect of multiple abiotic and biotic factors on Nup and NUE, which contribute greatly to the N cycle. Authors utilized a large-scale dataset using a model selection approach paired with a series of generalized mixed linear models. Interestingly, they found that Nup is primarily driven by abiotic factors such as N deposition, air temperature and precipitation, while NUE is driven by soil attributes such as pH, soil microbial stocks, and AMF presence. The authors also compared their results to eight different climate models included in the TRENDY models and claim that these models may be severely overestimating the Nup in ecosystems.

While I find some of their results insightful, particularly that Nup and NUE are driven by very different environmental factors, I am less convinced that their later result is as big as they claim. Mainly, that the TRENDY models are overestimating Nup to that large of degree when 5 out of the 7 models are very much in line with the estimates derived from this approach (Fig. S7). Thus, I do not think the claim that the TRENDY models is as large as the authors here are claiming. Rather, a more in-depth discussion about why there are two models that are so much higher would be much more useful.

Additionally, there is a lack of clarity regarding where this data came from. In the manuscript, it states that plot-level data was collected from 159 sites. But it unclear if these sites are or data were collected first-hand by the authors, if there were additional collaborators, if these are network sites, or if these was more of a meta-analysis approach in which this data was extracted from multiple published datasets. Also, in the manuscript itself, it does not mention how many datapoints total are included in this dataset, and that multiple points are coming from a single site. Where and how you attained the climatic data is very clear. However, I am confused about how you derived and interpret the AMF data, notably, what the percentages mean. Do they represent the maximum potential % colonization on roots or a likelihood of assassination? A little more clarity would ease interpretation of the data. Finally, looking at Fig. S8, you have severe data discrepancies in much of Africa, Russia, Australia, and the tropics in general. This leaves me very wary about your ability to scale up to the level that you do. It leads me to believe that the Nup differences are likely due to lack of data in those area rather than differences in how you model the biology. More discussion about this is needed in the interpretation of the results.

My last major concern is regarding the statistical analysis. While it is fairly clear how the analysis was performed, there are key details missing from the methods that are important. Importantly, the random effects structure (if any were included) is not discussed. Given that you have multiple points from a single site, data from a wide range of dates, and potentially some spatial autocorrelation, a mixed effect approach is most appropriate here. If there are random effects incorporated, this needs to be included into the methods – and if not, why? I think it would also be useful to present the variance inflation factor (VIF) scores here as well, as some variables may not be correlated, but overlap in the variance they explain. Also, a little more explanation on what "variable importance" would be useful for interpretation. Is it a derivative of $R^2$? Lastly, in terms of the upscaling approach, which I appreciate the amount of citations included here, I feel that there is still a lack of explanation for those who may be unfamiliar with this approach. Namely, if the validation ratio used here is standard, and what exactly is the parameter optimization approach

described in lines 460-461 (this can go in the supplement, but as written is it difficult to understand).

In terms of the writing, the flow of the manuscript is nice and easy to follow, However, I think a little more motivation and discussion is needed in the introduction. Specifically, the mechanisms in which the soil microbiota are important, the roles they play, and how the different types (AMF vs. microbes) influence the various processes. Then, the quick switch to a discussion about N deposition is abrupt and a little confusing, Overall, the first paragraph needs to be broken up into distinct topics and expanded. The results and discussion are structured nicely and in a way that makes sense. However, I think the methods could be more streamlined such as combining sections 4.1 and 4.2 into a single "data extraction" section. In general, the manuscript has grammar errors throughout that disrupt the flow of the reader.

---

## Author Response (AR1)

We are excited to provide the revised version of our manuscript "Nitrogen deposition and climate drive plant nitrogen uptake while soil factors drive nitrogen use efficiency in terrestrial ecosystems" for further consideration in ESD. In this updated version we have incorporated the feedback provided by the experts, which has increased the clarity and transparency of our work. In summary, the updates of this new version include:

- Further discussion in how our approach is only meant to be used at global scale, refraining from being used in more local purposes.
- A more detailed introduction, specially focusing on the soil processes than affect the N cycle
- A recalculation of our linear models aggregating our values by location, avoiding potential problems with spatial autocorrelation. This resulted in an updated Figure 1, 2 and S1.
- Further specification on our comparison between our Nup calculation and the Nup provided by the TRENDY models. We emphasize the spatial differences between the two, avoiding confusion with total values. That also resulted in a new version of figure S7.
- Inclusion of figure S10, which shows the correlation between Nup and NUE.
- Further details, clarifications and rephrasing in the methodology section.
- A grammatical check.

Following, we provide the responses point to point to the reviewer's comments.

REVIEWER 1:

This paper sought to quantify the effect of multiple abiotic and biotic factors on Nup and NUE, which contribute greatly to the N cycle. Authors utilized a large-scale dataset using a model selection approach paired with a series of generalized mixed linear models. Interestingly, they found that Nup is primarily driven by abiotic factors such as N deposition, air temperature and precipitation, while NUE is driven by soil attributes such as pH, soil microbial stocks, and AMF presence. The authors also compared their results to eight different climate models included in the TRENDY models and claim that these models may be severely overestimating the Nup in ecosystems.

While I find some of their results insightful, particularly that Nup and NUE are driven by very different environmental factors, I am less convinced that their later result is as big as they claim. Mainly, that the TRENDY models are overestimating Nup to that large of degree when 5 out of the 7 models are very much in line with the estimates derived from this approach (Fig. S7). Thus, I do not think the claim that the TRENDY models is as large as the authors here are claiming. Rather, a more in-depth discussion about why there are two models that are so much higher would be much more useful.

We want to thank the reviewer for the time devoted to read and provide excellent feedback to our manuscript. We find the provided comments very useful to improve the transparency, clarity and flow of our manuscript, which we incorporate in the new version of our manuscript.

Thanks to your comment, we realized about the potential misunderstanding between global yearly differences and spatial differences between the TRENDY models and our estimations. To avoid giving the wrong impression we reformulate the wording and modify figure S7 as follows:

We now expand the methodology section better explaining how the comparison has been done as such (Line 614-618): "We calculated the yearly mean Nup from 1984 to 2022 for each model, and the average of all of them combined, to obtain a final yearly value. Then, we calculated the difference between each model included in the TRENDY ensemble and our Nup estimations.

After, we averaged all the fNup values in a unique spatial explicit representation and compared it with our estimations. We also calculated the latitudinal mean of the difference to achieve a latitudinal profile and calculated the overall mean".

We now include the average value of all the TRENDY models combined in a new version of Fig. S7 as follows, to support your clever observation:

[Figure]

We also rephrase part of the 2.6 Discrepancies between Nup map and TRENDY section as such to align with your comment (Line 374-380):

**2.6 Latitudinal discrepancies between Nup map and TRENDY**
LPX-Bern and CLM5.0, models included in the TRENDY ensemble, projected Nup values significantly above our estimations, being 1471 and 1454 Tg N/yr respectively (Fig. S7). Even though the average of all TRENDY models falls within our Nup confidence range differences become relevant through space. In tropical and northern latitudes, TRENDY models projected higher values than our estimations. On the other hand, TRENDY models projected lower Nup values in southern latitudes, western Asia and at the Rocky Mountains."

This way we emphasize how the differences we highlight are on the spatially explicit representation rather than in the total average number, while identifying the models that overestimate the Nup values.

Additionally, there is a lack of clarity regarding where this data came from. In the manuscript, it states that plot-level data was collected from 159 sites. But it unclear if these sites are or data were collected first-hand by the authors, if there were additional collaborators, if these are network sites, or if these was more of a meta-analysis approach in which this data was extracted from multiple published datasets. Also, in the manuscript itself, it does not mention how many datapoints total are included in this dataset, and that multiple points are coming from a single site. Where and how you attained the climatic data is very clear. However, I am confused about how you derived and interpret the AMF data, notably, what the percentages mean. Do they represent the maximum potential % colonization on roots or a likelihood of assassination? A little more clarity would ease interpretation of the data. Finally, looking at Fig. S8, you have severe data discrepancies in much of Africa, Russia, Australia, and the tropics in general. This leaves me very wary about your ability to scale up to the level that you do. It leads me to believe that the Nup differences are likely due to lack of data in those area rather than differences in how you model the biology. More discussion about this is needed in the interpretation of the results.

Thank you for bringing up this lack of clarity in our methodology. We further describe and clarify your questions with pleasure in the new version. The data gathering process follows a meta-analysis approach (the consulted bibliography can be found in Table S1) and the gathered information came from 129 different locations. We now include both statements in the methodology section of the new version as follows (Line 453-455): "We gathered 159 field plot data representing 129 different sites in natural conditions coming from published data or repositories (Table S1). The data included information on the dominant species...".

Regarding the AMF data we used the data from Soudzilovskaia et al., 2019 and 2020 incorporating their approach in our analysis. The AM% is the percentage of root colonization by AMF at plot level. Our inference process is better detailed in the new version of the manuscript as such (Line 487-492): "We also identified the potential mycorrhizal association from the dominant species (% of colonization) based on Soudzilovskaia et al. 2020, and categorized it into 0, 50, or 100 arbuscular mycorrhizal (AM) percentages, since AM is the most abundant and common throughout the globe. When dominant species were not provided, we extracted the AM% of colonization based on the AM map of Soudzilovskaia et al. 2019 and the coordinates of our samples."

We agree that the quality of upscaling efforts is tied to data availability and that it exists a systematic underrepresentation of the global south. The exact same concern you are arising is the one we wanted to tackle when we wrote the section 2.7 about representativity and future research. For example (Line 392-417): "An inherent challenge in ecological studies of this scale is to ensure the global representativeness of the dataset, since there are systematic geographical sampling biases underrepresenting the global south. In this study, the 28 % of the data comes from areas below 15º latitude, outside the US, Europe, or China (Fig. S8). When accounting for ecosystems representativity, the Whittaker diagram shows we have a representation of all the biomes (Fig. S9), showing the lowest representativity on subtropical desert, tundra, and temperate rainforest. Nonetheless, we acknowledge that calculations based on empirical data, especially when a portion of the data have undergone a gap-filling process, can still have biases associated with sampling and the upscaling process, which are mainly defined by the more represented biomes of the observations. Still, we believe that calibrating and cross-checking models built over mathematical assumptions with field measurements is necessary to better root models to reality."

Nonetheless, to prevent further misunderstandings about our findings we will incorporate this new paragraph in the new version of the manuscript (Line 419-423): "Global scale approaches such as this one, are intended to provide broad quantifications of planetary processes assuming a scale-precision compromise. Because of that, we do not advise using our approach to seek fine-scale precision, since our results may overlook local particularities and differences between forest type, species distribution, or management practices." as well as this further recommendation for further research (Line 429-430): "We also encourage empirical studies sampling underrepresented biomes, specially grom the global south."

My last major concern is regarding the statistical analysis. While it is fairly clear how the analysis was performed, there are key details missing from the methods that are important. Importantly, the random effects structure (if any were included) is not discussed. Given that you have multiple points from a single site, data from a wide range of dates, and potentially some spatial autocorrelation, a mixed effect approach is most appropriate here. If there are random effects incorporated, this needs to be included into the methods – and if not, why?

This is indeed an excellent point that needs to be furtherly explored. From all the sites we include in our study, only 15 of them had repetitions (Data that falls in the same pixel accounting for a 1km spatial resolution). Despite being only a few, the site repetition could be a vulnerability to the data independency required for the GLM's used here. Since our study aims to calculate average Nup and NUE per unit of surface we approached the repetitions aggregating the sampled values that fall into the same pixel. We have recalculated our linear models with the data aggregated by site and the important variables and outputs of the models remained the same, ensuring the robustness of our glm's. Please, see the R outputs attached (without and with aggregation respectively).

```
Coefficients:
              Estimate Std. Error t value Pr(>|t|)
(Intercept)  2.200e-02  1.632e-03  13.486  < 2e-16 ***
MAP_combi   -1.821e-06  7.010e-07  -2.598   0.0103 *
MAT_combi   -3.066e-04  7.280e-05  -4.212 4.29e-05 ***
noy         -1.960e-04  4.639e-05  -4.226 4.04e-05 ***
```

```
              Estimate Std. Error t value Pr(>|t|)
(Intercept)  2.190e-02  1.921e-03  11.401  < 2e-16 ***
MAP_combi   -2.015e-06  7.221e-07  -2.791  0.00608 **
MAT_combi   -2.679e-04  8.203e-05  -3.265  0.00141 **
noy         -2.099e-04  5.862e-05  -3.580  0.00049 ***
```

In the new version of the manuscript, we are recalculating figure 1, 2 and S1 accordingly to match with the new approach.

I think it would also be useful to present the variance inflation factor (VIF) scores here as well, as some variables may not be correlated, but overlap in the variance they explain.

Following your advice we also performed a VIF analysis as a safety check. We calculated the VIF for the aggregated glm model and all the variables fell under the safe value of 4.

| Alt_combi | AM_combi | MAT_combi | mic_N | noy | woodiness | N_stocks_combi | MAP_combi |
|---|---|---|---|---|---|---|---|
| 1.477350 | 2.136810 | 3.745478 | 1.491679 | 1.720187 | 1.257470 | 1.757075 | 3.277356 |
| Soil_ph_combi | | | | | | | |
| 1.953054 | | | | | | | |

| MAT_combi | AM_combi | MAP_combi | mic_N | Alt_combi | N_stocks_combi | Soil_ph_combi |
|---|---|---|---|---|---|---|
| 3.296120 | 1.180451 | 3.567814 | 2.480000 | 1.291476 | 1.732917 | 3.148289 |
| woodiness | noy | | | | | |
| 1.061319 | 1.498485 | | | | | |

To make this clearer to the reader we incorporate this information in the methodology section as follows (Line 543 – 545): "We also calculated the Variance Inflation Factor (VIF) with the *vif* function of the car R package (Fox and Weisberg, 2019) of the aggregated model to validate VIF's lower than 4."

Also, a little more explanation on what "variable importance" would be useful for interpretation. Is it a derivative of $R_2$?

Thank you for your comment. We will include this sentence in the methodology section to better explain the variable importance (Line 547 – 549): "We calculated the variable importance using the function *sw* on the MuMIN R package (Barton, 2023), which is a standard method based on Akaike weights (Giam and Olden, 2016)."

For further clarification we also include this statement in the main section to deepen in the variable importance meaning (Line 212 – 215): "Despite climatic variables such as MAT and MAP not appearing as important variables explaining NUE, they are to some extent represented in the soil variables. As shown in Table S2, soil variables are not independent of climatic variables since they have some degree of correlation."

Lastly, in terms of the upscaling approach, which I appreciate the amount of citations included here, I feel that there is still a lack of explanation for those who may be unfamiliar with this approach. Namely, if the validation ratio used here is standard, and what exactly is the

parameter optimization approach described in lines 460-461 (this can go in the supplement, but as written is it difficult to understand).

We included further details in the section 4.6 of our manuscript such as (Line 554 – 556): "To upscale Nup and NUE to global grasslands and woody vegetation, we used extreme gradient boosting (XGBoost) models splitting the database into train, test, and validation using a standard ratio of 70:20:10, respectively (Lever et al., 2016)"; (line 559 – 561): "XGBoost is a non-parametric model particularly indicated for high performance in sample sizes above 100 data points, overcoming potential problems of autocorrelation and optimizing predictive power" or (line 578-580) "We optimized the parameters based on performance (prediction $R^2$) at a maximum depth of 6, minimum child weight of 1, and eta of 0.3, which are generally standard values".

In terms of the writing, the flow of the manuscript is nice and easy to follow, However, I think a little more motivation and discussion is needed in the introduction. Specifically, the mechanisms in which the soil microbiota are important, the roles they play, and how the different types (AMF vs. microbes) influence the various processes. Then, the quick switch to a discussion about N deposition is abrupt and a little confusing, Overall, the first paragraph needs to be broken up into distinct topics and expanded.

We strongly agree. In the new version of the manuscript we separate the first paragraph in two and expand the motivation about the soil community as follows (line 113 – 124): "The soil community (i.e. microbes and fungi) play a crucial role on global biogeochemical cycles governing processes such and N fixation, nitrification, denitrification and general organic matter and nutrient turnover (Aber et al., 2001; Sinsabaugh et al., 2002; Sinsabaugh et al., 2008; Crowther et al., 2019; Delgado-Baquerizo et al., 2020). In turn, the soil community can also act as a buffer in case of nutrient excess (Wall et al., 2015) or contribute to nutrient foraging in case of nutrient deficiencies (Chen et al., 2018), shaping ecosystem functioning (Bardgett and van der Putten, 2014). Therefore, the soil community is expected to substantially interfere in the plant Nup and NUE. N deposition is another agent relevant for global N cycles, which has increased from ~30 to ~80 Tg N/year worldwide since 1850 (Kanakidou et al., 2016) with an associated increase on N availability (Elser et al., 2010; Battye et al., 2017; Peñuelas et al., 2020). Consequently, reliable quantifications of plant Nup and NUE need to include climatic factors as well as soil biotic factors and N deposition."

The results and discussion are structured nicely and in a way that makes sense. However, I think the methods could be more streamlined such as combining sections 4.1 and 4.2 into a single "data extraction" section. In general, the manuscript has grammar errors throughout that disrupt the flow of the reader.

Thank you. We now combine section 4.1 and 4.2 into one section as well as section 4.3 and 4.4. In the new version we have a total of 5 subsections instead of 7. We also performed a general grammar check to avoid disruption and increase flow.

REVIEWER 2:

"Nitrogen deposition and climate drive plant nitrogen uptake while soil factors drive nitrogen use efficiency in terrestrial ecosystems" by Vallicrosa et al. presents an analysis of ground-based estimates of nitrogen uptake and nitrogen use efficiency using NPP of foliage, wood (if present), and fine roots along with nitrogen concentration data for each of those tissues. The data span all the biomes, and the authors pair those measurements with plausible drivers. Climatic variables and N deposition appear to drive nitrogen uptake, whereas soil biological variables appear to drive nitrogen use efficiency. Total soil nitrogen was not predictive of either response.

I am impressed by the data collection, which seems reasonable to me. I am satisfied by the equations presented (subject to some small points below) and the measures the authors used to fill in missing data. I think the results are believable and interesting and useful.

We thank the reviewer for the very positive comments and time devoted to improving our manuscript, providing deep thinking and arising excellent points. By incorporating the requested arrangements, we believe that the clarity and fairness of our study will be substantially improved. We deeply appreciate your rigorous but simultaneously kind approach in your review.

I have only two major requests. First, I can see that the Nup and NUE global predictions are not simply negatives of each other, but I can also see that there is a negative correlation between Nup and NUE. Would it be reasonable to provide a plot of these two key variables against each other? I attempted this with the supplemental data, and there was clearly a "wedge": high Nup paired with low NUE and high NUE paired with low Nup. But intermediate values show a lot of variation. I assume the different predictors explain a lot of that variation. OK, here's my request: please explain why a reader who casually looked at figure 3 would be wrong to conclude that climatic drives are affecting both Nup and NUE in opposite directions. It just seems weird that none of the climatic drivers rise to the top of the NUE analysis, especially when I look at figure 3b. Put another way, the interesting result that Nup is driven by climate and NUE is driven by biology warrants deeper discussion, along with ways that the "biology" might also be correlated with climate (or not).

This is an excellent point, thank you for pointing that out. What figure 1a and 2a are disclosing is what are the most important variables to explain Nup and NUE respectively. In other words, which are the variables that explain a higher portion of the variability of the dependent variable (Nup or NUE). Nonetheless, the fact that a variable it is not disclosed as important doesn't make it uncorrelated, as you mention in your comment. As we report in the table S2 and line 442 we see some level of correlation between soil and climatic variables being 0.52 the highest one, between MAT and AM presence. Thus, aligning with your point of view, soil variables show to embed climatic factors to some extent since they are not totally independent ecologically. We see how this can create confusion to our potential readers so we now better clarify this aspect in

the section 2.1as follows (line 212-215): "Despite climatic variables such as MAT and MAP not appearing as important variables explaining NUE, they are to some extent represented in the soil variables. As shown in Table S2, soil variables are not independent of climatic variables since they have some degree of correlation". We also removed sentences that could give opposite impressions to avoid confusion in the abstract and the main body of the manuscript.

In addition, we are now including a supplementary plot (Fig. S10) showing the relationship between Nup and NUE as follows in the methodology section.

[Figure]

My second major request is for some caveats and discussion about the global nature of your conclusions. I work in temperate forests – the drivers of Nup and NUE within this much smaller set of points may or may not match the drivers you found globally. I think that is worth saying loudly (i.e. in the abstract as well as the discussion). In other words, I think you have found great reasons why Nup and NUE will differ between, say, a boreal forest and a tropical forest, but your analysis doesn't say anything about why, say, a pine forest will have a different Nup and NUE than a nearby maple-basswood forest (right?). I would hate for folks who are new to this area or who are only thinking about this superficially to imagine that the strong results you found globally also apply locally. (They may, but your analysis can't address whether they do or don't.)

This is indeed a caveat of global approaches that needs to be responsibly acknowledged. We believe our 2.7 section is the best to include such discussion. There, we will incorporate further discussion as such (line 419-423): "Global scale approaches such as this one, are intended to provide broad quantifications of planetary processes assuming a scale-precision compromise. Because of that, we do not advice to use our approach to seek fine scale precision, since our results may overlook local particularities and differences between forest type, species distribution or management practices". We also emphasize in our abstract that our approach is valid for global scale quantifications, line 64.

Minor things:
Line 51: "we used ground-based observations..." can you say something more about the nature of these measurements?

With pleasure. We now include: "...we used ground-based observations across 159 field experiments (including above and belowground information)..."

Line 53-54: do you mean "mean temperature and precipitation"?

Yes. We modify it as mentioned.

Line 165: Looks like the dashed line is at 0.75, not 0.8.

You are right, thank you. We now modify the footnote accordingly.

Line 165 (and 180): I don't know what "The model preudoR2 was 0.349" means (oh, is it the R2 for the model? This can be clarified.)

Yes, it is. We realize the typo now and we changed it to GLM pseudoR$^2$ instead.

Line 173: Is "contraposition" the correct word here? "In contrast" would work nicely I think

It is changed as you recommend.

Line 244, 275: I would recommend rounding to whole numbers – three decimal places seems like a mismatch with the uncertainty in these numbers.

It is changed to 842 ± 236 Tg of N and 110 ± 19 kg C per kg N in the text as you suggest.

Line 263: I'm not sure what you mean by, "Nonetheless, these efforts do not translate yet on low N deposition effects in natural woodlands and grasslands."

We agree it is not straight forward. We rephrase it to "Nonetheless, these reductions do not translate yet into how N deposition effects natural woodlands and grasslands, showing to still be the main driver for plant Nup."

Line 269: I understand this sentence, but I think it can be rephrased more clearly

It isrephrased as such: "Consequently, this extra input of N coming from N deposition is not being captured by biomass and enhances the N leaching associated with eutrophication, acidification, loss of biodiversity, and N$_2$O emissions (Aber et al., 1989; Gundersen et al., 1998; Bobbink et al., 2010) exacerbating environmental problems."

Line 328: "truthfully" seems like an odd word to use here. Perhaps "more accurate"?

We incorporate it as suggested.

Line 418: Seems strange to call this "GrossNup" since they actually aren't taking up the portion that they were able to retranslocate. What you are calling "NetNup" is what they are actually taking up. I'd suggest simplifying this and avoiding confusion by embedding the retranslocation within a single equation for "Nup".

We rearrange it in one single equation as such:

$$Nup = (NPPleaves * Nleaves + NPPstem * Nstem + NPProots * Nroots) - (NPPleaves * Nleaves - NPPlitter * Nlitter)$$

Line 424: these are fractions, not percentages, right?

The most common units to express N in plant tissues is percentage in dry weight (%) or mg of N per g of dry biomass (mg/g). In this case we chose %. We changed it to % in dry weight to make it clearer.

Line 423, 434: shouldn't this be kg biomass/ha/yr?

Indeed. It is modified accordingly. Thank you for pointing this out.

Line 435: These units don't seem to be working out

We now express it the same way as for the other formula.

---

## Author Response (AR2)

I think this version has addressed my earlier comments and will be a useful contribution to the literature. Stated humbly as someone who can only speak and write a single language: the English in this revision seems to need more thorough editing prior to publication. There are many distracting grammatical errors and homophone errors (e.g. affect vs effect).

We thank the reviewer for the time devoted in assessing our manuscript and providing valuable feedback. We appreciate their positive comments that have significantly increase the quality of our work. Following their advice, now we have undergone a deep revisiting of the english writing, increasing the reading experience of the manuscript.